# Risk of Pre-Malignancy or Malignancy in Postmenopausal Endometrial Polyps: A CHAID Decision Tree Analysis

**DOI:** 10.3390/diagnostics11061094

**Published:** 2021-06-15

**Authors:** Michael Wong, Nikolaos Thanatsis, Federica Nardelli, Tejal Amin, Davor Jurkovic

**Affiliations:** 1Institute for Women’s Health, University College London Hospitals, London NW1 2BU, UK; michael.wong3@nhs.net (M.W.); nikolaos.thanatsis@nhs.net (N.T.); tejal.amin@nhs.net (T.A.); 2Department of Women’s and Children’s Health, Catholic University of Sacred Heart, 1, 00168 Rome, Italy; nardelli.fede@gmail.com

**Keywords:** endometrial polyp, postmenopausal women, cancer risk, ultrasound, endometrial cancer, endometrial hyperplasia

## Abstract

Background and aims: Postmenopausal endometrial polyps are commonly managed by surgical resection; however, expectant management may be considered for some women due to the presence of medical co-morbidities, failed hysteroscopies or patient’s preference. This study aimed to identify patient characteristics and ultrasound morphological features of polyps that could aid in the prediction of underlying pre-malignancy or malignancy in postmenopausal polyps. Methods: Women with consecutive postmenopausal polyps diagnosed on ultrasound and removed surgically were recruited between October 2015 to October 2018 prospectively. Polyps were defined on ultrasound as focal lesions with a regular outline, surrounded by normal endometrium. On Doppler examination, there was either a single feeder vessel or no detectable vascularity. Polyps were classified histologically as benign (including hyperplasia without atypia), pre-malignant (atypical hyperplasia), or malignant. A Chi-squared automatic interaction detection (CHAID) decision tree analysis was performed with a range of demographic, clinical, and ultrasound variables as independent, and the presence of pre-malignancy or malignancy in polyps as dependent variables. A 10-fold cross-validation method was used to estimate the model’s misclassification risk. Results: There were 240 women included, 181 of whom presented with postmenopausal bleeding. Their median age was 60 (range of 45–94); 18/240 (7.5%) women were diagnosed with pre-malignant or malignant polyps. In our decision tree model, the polyp mean diameter (≤13 mm or >13 mm) on ultrasound was the most important predictor of pre-malignancy or malignancy. If the tree was allowed to grow, the patient’s body mass index (BMI) and cystic/solid appearance of the polyp classified women further into low-risk (≤5%), intermediate-risk (>5%–≤20%), or high-risk (>20%) groups. Conclusions: Our decision tree model may serve as a guide to counsel women on the benefits and risks of surgery for postmenopausal endometrial polyps. It may also assist clinicians in prioritizing women for surgery according to their risk of malignancy.

## 1. Introduction

Endometrial polyps are common uterine lesions that are present in 5.8% of premenopausal and 11.8% of postmenopausal women [1]. They are a heterogeneous group of lesions that could be divided cytogenetically into at least four different subgroups with various clonal chromosomal rearrangements [2]. The pathophysiology and natural history of endometrial polyps are unclear, as many are found in asymptomatic women. Some cohort studies reported that the growth rate of polyps cannot be predicted by the clinical history or patient’s demographics, but a small proportion of polyps may regress spontaneously [3].

Currently, there is no consensus on the management of postmenopausal endometrial polyps. Some advocate for the removal of all postmenopausal polyps, as pre-malignancy (atypical hyperplasia) or malignancy cannot be excluded [4], while others may consider expectant management, especially for asymptomatic women, as the risk of pre-malignancy or malignancy is low [5]. Furthermore, expectant management may also be considered in women with significant medical co-morbidities or a failed hysteroscopy, where the risk of surgery outweighs the risk of malignancy.

According to Scott [6], a malignant endometrial polyp is defined as having malignant cells confined to one surface of the polyp and the endometrium around the base of the polyp shows no changes of malignancy. In a recent meta-analysis, the estimated risk of pre-malignancy or malignancy in postmenopausal polyps was 4.9% [7]. Risk factors for pre-malignancy or malignancy include older age, obesity, hypertension, diabetes mellitus, a history of abnormal uterine bleeding, or tamoxifen use [8].

Clinically, it is important to offer all women an individualised discussion about their management options. In a study on women’s preferences, 59% of women with postmenopausal bleeding would like 100% certainty that malignancy has been ruled out, while 36% of women would accept expectant management if the risk of malignancy is ≤5%, and a small proportion of women (5%) may choose expectant management even if the risk of malignancy is >5% [9].

We hypothesized that the patient’s clinical characteristics and ultrasound morphological features of the polyp may help improve the risk prediction of pre-malignancy or malignancy in postmenopausal polyps.

This study aimed to carry out a decision tree analysis to identify the predictive patient characteristics and ultrasound features of polyps for pre-malignancy or malignancy in postmenopausal polyps.

## 2. Materials and Methods

### 2.1. Study Design

This was a prospective study conducted between October 2015 and September 2018 at a general gynaecology outpatient clinic of a university teaching hospital. All patients were referred via their general practitioners. We included consecutive postmenopausal women who were diagnosed with endometrial polyps on transvaginal ultrasound examination and underwent hysteroscopic polypectomy or hysterectomy within 3 months of the ultrasound assessment. Menopause was defined as women aged 45 or above with at least a 12-month history of amenorrhoea. We excluded women who were on tamoxifen or with a known history of endometrial hyperplasia or malignancy.

All ultrasound examinations were carried out by a Level-II operator [10] with an ultrasound system equipped with a 4–9 MHz transvaginal probe (Voluson E8, GE Healthcare Ultrasound, Milwaukee, WI, USA). The diagnosis of an endometrial polyp on ultrasound was made in accordance with the IETA consensus when there was a well-defined focal lesion with a regular outline within the endometrial cavity [11]. The surrounding endometrium appeared morphologically normal. On Doppler ultrasound, there was either a single feeder vessel or there was no detectable vascularity [12] (Figure 1). Each polyp was measured in 3 perpendicular planes (d1, d2, d3) in the longitudinal and transverse views of the uterus. The polyp mean diameter (dm) was calculated from these measurements (i.e., dm = (d1 + d2 + d3)/3) and expressed in millimetres. If multiple polyps were present, only the largest polyp was included in our final analysis. Additionally, each polyp was assessed for the presence of intralesional cystic spaces (Figure 2). Polyps were described as cystic if they contained any intralesional cystic spaces, or as solid if they did not have any visible cystic spaces. In cases where the endometrium could not be assessed adequately or if the diagnosis of a polyp was uncertain, saline infusion sonography (SIS) was performed. Endometrial lesions with an irregular surface (with or without SIS), a multi-vessel vascular pattern on Doppler, or abnormal adjacent endometrium were diagnosed as suspected endometrial cancers [13], and they were excluded from the study.

For each patient, we recorded their clinical risk factors for endometrial hyperplasia or malignancy, which included age, body mass index (BMI), parity, use of hormone replacement therapy (HRT), history of hypertension, and diabetes mellitus [14,15,16,17].

Following hysteroscopic polypectomy or hysterectomy, all surgical specimens were examined by pathologists who were blinded to the ultrasound assessments. The final histological diagnosis was used as our gold standard. We divided women into 2 categories for our analysis: (1) benign polyps, which included polyps with hyperplasia but no evidence of atypia, and (2) polyps with atypical hyperplasia or malignancy.

### 2.2. Construction of the CHAID Decision Tree

We used the Chi-squared automatic interaction detection (CHAID) algorithm [18,19] to perform our decision tree analysis, with the dependent variable defined as the presence or absence of atypical hyperplasia or malignancy in endometrial polyps. Our independent variables were the patient’s age, BMI, parity, history of hypertension, diabetes mellitus, use of HRT, number of polyps, polyp mean diameter, presence or absence of a feeder vessel to the polyp, and whether the polyp appeared cystic or solid on ultrasound. The CHAID algorithm is a non-parametric procedure and, therefore, it required no assumptions to be made of the underlying data. Multiple 2 × 2 contingency tables between the dependent variable and each independent variable were created; the most significant independent variable in a Chi-square test was then selected to branch out the decision tree. The categories of each independent variable were merged if they were not significantly different from the dependent variable [20]. The decision tree was set to have a maximum of 3 levels, a minimum of 20 cases in each parent node, and any given split should not generate a child node with fewer than 10 cases; the significance level (α_merge_, α_split_, and *p*-value) was set at ≤0.05. The resulting subgroups created by the decision tree model were divided into 3 classification groups according to the risk of pre-malignancy or malignancy in endometrial polyps: low-risk (≤5%), intermediate-risk (>5% to ≤20%) or high-risk (>20%).

Descriptive statistical methods were used to describe the study population. Comparisons of the population characteristics were unpaired, and all tests of significance were two-tailed. Continuous variables were compared using the Mann–Whitney U test. Categorical variables were compared using Pearson’s Chi-square or Fisher’s exact test. A significance level of <0.05 was used without multiple comparison adjustment. Statistical analyses were performed using IBM SPSS Statistics for Windows, version 25 (IBM Corp., Armonk, NY, USA).

### 2.3. Decision Tree Validation

We used the 10-fold cross-validation method to internally validate our decision tree model [20]. In this method, the original study cohort is randomly partitioned into 10 subsets of equal sizes. Of the subsets, 1 is used as the validation set, while the other 9 are used as the training set. The cross-validation process is repeated 10 times, in which each of the 10 subsets is used only once as the validation set. The average value of the 10 results from the folds is estimated as the misclassification risk value.

## 3. Results

During the study period, 1686 postmenopausal women underwent transvaginal ultrasound examination (Figure 3). Of the 1534 eligible women, 886 women (58%) presented with postmenopausal bleeding. For the remaining 648 women, who did not present with abnormal bleeding, their indications for ultrasound examination are summarised in Table 1. A total of 308 endometrial polyps were diagnosed on ultrasound. The proportion of women diagnosed with polyps is similar to those presenting with or without postmenopausal bleeding (192/886 (22%) vs. 116/648 (18%); X^2^ (1, *N* = 1534) = 3.3, *p* = 0.07).

All women with endometrial polyps were offered surgery; however, 68/308 (22%) were managed expectantly due to the woman’s preference, the presence of medical co-morbidities, or a failed hysteroscopy. In women with postmenopausal bleeding, 11/192 (6%) polyps were managed expectantly, compared to 57/116 (49%) in women without postmenopausal bleeding. On univariate analysis, expectantly managed asymptomatic women were significantly older (median of 70 (IQR 62–78) vs. 63 (IQR 56–73) (*p* = 0.01)) and they were more likely to have a single polyp ((98.2% vs. 83.1%) (*p* = 0.01)) of a smaller size (median of 10 mm (IQR 7.0–12.7) vs. 13.3 mm (IQR 9–17) (*p* = 0.01)), with no detectable vascularity on Doppler examination ((66.7% vs. 45.8%) (*p* = 0.03)), when compared to asymptomatic women who were managed surgically.

A final 240 women with polyps who underwent surgery in the form of hysteroscopic polypectomy or hysterectomy were included in our decision tree analysis. Patient characteristics are summarised in Table 2. There were 5/240 (2%, 95% CI 0.7–4.8) polyps with hyperplasia without atypia, 8/240 (3%, 95% CI 1.4–6.1) with atypical hyperplasia, and 10/240 (4%, 95% CI 2.0–7.5) were found to harbour malignancy. Overall, the prevalence of pre-malignancy or malignancy was 18/240 (8%, 95% CI 4.5–11.6). Among the malignant polyps, six were of endometrioid histological subtype and the other four were of serous histological subtype. On univariate analysis, pre-malignant or malignant polyps were significantly larger, more likely to appear solid rather than cystic, and less likely to appear avascular on colour Doppler imaging (Table 3). The number of polyps on ultrasound did not appear to be associated with pre-malignancy or malignancy.

### Decision Tree Analysis and Internal Validation

In our decision tree analysis, the three most significant predictive variables for pre-malignant or malignant polyps were polyp size, women’s BMI, and whether the polyp appeared cystic or solid on ultrasound. The model concluded with a total of five subgroups, which divided women into low-risk, intermediate-risk, or high-risk for pre-malignancy or malignancy (Figure 4).

The polyp mean diameter was selected as the first splitting variable in our model. For women with a polyp mean diameter of ≤13 mm, their risk of pre-malignancy or malignancy was 4/166 (2%, 95% CI 0.7–6.1); however, in polyps >13 mm, their risk was 14/74 (19%, 95% CI 10.7–29.7).

Among women with a polyp mean diameter of >13 mm, whether the polyp appeared cystic or solid was selected as the second splitting variable. Polyps that appeared cystic had a 1/37 (3%, 95% CI 0.1–14.2) risk of pre-malignancy or malignancy, and they were classified as low risk. Polyps that appeared solid were further divided with the woman’s BMI as the third splitting variable. In women with a BMI > 28.2, their risk of pre-malignancy or malignancy was 9/14 (64%, 95% CI 39.2–89.4) and they were classified as high-risk; whereas those with a BMI ≤ 28.2 had a risk of 4/23 (17%, 95% CI 1.9–32.9) and they were classified as intermediate-risk.

The risk of pre-malignancy or malignancy in women with a polyp mean diameter of ≤13 mm was generally low; however, our model further divided these women according to their BMI as the second splitting variable. In those with a BMI > 28.4, their risk of pre-malignancy or malignancy was 4/48 (8%, 95% CI 0.5–16.2), and they were classified as intermediate-risk, while there were no cases of pre-malignancy or malignancy in women with a BMI ≤ 28.4, and they were classified as low risk.

Using our decision tree model to counsel women in the high-risk and intermediate-risk groups for surgery, where the risk of pre-malignancy or malignancy is >5%, results in 85/240 (35%, 95% CI 29–42) women having hysteroscopic resection of polyps. The overall accuracy of our model for correctly identifying women with premalignant or malignant polyps was 94%. For internal validation, a misclassification risk of 8% ± 1.8% (standard error) was calculated using the 10-fold cross-validation method. This result means that our model may correctly diagnose 92% (95% CI 86.0–97.4) of premalignant or malignant polyps; the corresponding sensitivities and specificities of our model were 94.4% (95% CI 72.7–99.9) and 69.4% (95% C.I. 62.9–75.4).

## 4. Discussion

In this study, we carried out a decision tree analysis to classify postmenopausal polyps into low-risk (≤5%), intermediate-risk (>5% to ≤20%), or high-risk (>20%) for pre-malignancy or malignancy. We found that the polyp’s size, patient’s BMI, and intralesional cystic spaces on ultrasound were the best discriminators between benign and premalignant/malignant polyps. There were no cases of pre-malignancy or malignancy in women with a BMI ≤28.4 presenting with polyps measuring ≤13 mm in mean diameter. On the other hand, two-thirds of women with a BMI >28.2 and solid polyps >13 mm in size were diagnosed with pre-malignancy or malignancy on histological examination.

As our model advocates for prioritized polyp resection for women in the high-risk or intermediate-risk groups, only 1/18 malignant polyps was incorrectly classified into the low-risk group. This misdiagnosis occurred in a 58-year-old woman who had a BMI of 34.3, and an endometrial polyp measuring at 22 mm in mean diameter. This case highlights that although the presence of intralesional cystic spaces was useful in predicting benign polyps in women with a polyp >13 mm, false-negative diagnoses can occur in a small number of premalignant/malignant polyps that appear cystic rather than solid.

Few prospective studies have reported on the prevalence of malignancy in endometrial polyps, nonetheless, the prevalence of 4% in our study is in keeping with the pooled estimate of 5.1% (95% CI 3.5–6.8) in a recent meta-analysis [7].

Traditionally, larger polyps are thought to have an increased risk of malignancy, which has been confirmed by our findings [21]. We found three previous studies that have assessed polyp size in postmenopausal women and the risk of pre-malignancy or malignancy [22,23,24]. All of these studies reported a larger polyp size was associated with an increased risk of pre-malignancy or malignancy. Various cut-offs were recommended by these studies, which included ≥18 mm, ≥19.5 mm, and ≥30 mm, respectively. However, unfortunately, a systematic review by Lee et al. [25] concluded that a meta-analysis was not possible due to the different measurement units used by various studies.

Obesity (BMI ≥ 30) is a well-known independent risk factor for endometrial hyperplasia and type 1 endometrial cancer. A recent study has also found that obesity is significantly associated with endometrial polyps in postmenopausal women [26]. This is in line with immunohistochemical studies that showed that obese postmenopausal women have a higher proportion of oestrogen receptor (ER) positive cells in the glands and stroma of their polyps, which is in contrast to the low ER expression in atrophic endometrial cells, suggesting steroid receptors have a crucial role in the pathophysiology of postmenopausal polyps [27,28]. In a meta-analysis of 3612 women, obesity was also significantly associated with an increased risk of pre-malignancy or malignancy in polyps [8]. Using a cut-off of BMI ≥ 32.5, Ghoubara et al. [29] reported that the sensitivity and specificity for hyperplasia or malignancy in postmenopausal polyps were 77% and 52%, respectively. In our study, we confirmed that a raised BMI was useful to identify subgroups of women with an increased risk of premalignant/malignant polyps.

Intralesional cystic spaces on ultrasound are thought to represent the dilated glands of endometrial polyps histologically and they could be lined by atrophic, inactive, or proliferative endometrium. In a study of focal endometrial lesions in premenopausal and postmenopausal women, 58.6% of the benign polyps had intralesional cystic spaces [30]. In postmenopausal polyps, the prevalence of intralesional cystic spaces was even higher at 72.4% [31]. Goldberg et al. [32] compared the greyscale morphological features of benign and malignant endometrial lesions and found that intralesional cystic spaces were more common in benign than malignant lesions (62% vs. 6%, respectively). Our results are in line with previous studies that reported that intralesional cystic spaces can be used to identify a subgroup of polyps at low risk of pre-malignancy or malignancy.

An endometrial polyp is considered a cause of abnormal uterine bleeding. Some studies have suggested that aberrant angiogenesis in the polyp plays a significant role; it causes the endometrial vessels over the polyp to dilate and become fragile, and, therefore, they are more prone to bleeding [33]. However, it is less certain whether polyps in women with symptoms of abnormal uterine bleeding have an increased risk of malignancy, as some suggested that the seemingly higher prevalence of pre-malignancy or malignancy amongst symptomatic women could be due to detection bias [34]. In our study, we found that symptoms of postmenopausal bleeding did not improve the risk prediction of pre-malignancy or malignancy in polyps when other patient characteristics and polyp morphological features were taken into account.

There were some limitations to our study. Firstly, a significant proportion of polyps (49%) in women without postmenopausal bleeding were managed expectantly and excluded from our analysis. Therefore, we cannot rule out the possibility of a selection bias, given that expectantly managed polyps were smaller in size and less likely to have detectable vascularity on Doppler examination. The risk of pre-malignancy or malignancy in incidentally diagnosed polyps could therefore be lower than those reported in our study. Secondly, the polyp’s size measurement and subjective assessment of intralesional cystic spaces in polyps could be affected by intra- and inter-rater variability. Thirdly, our decision tree model needs to be externally validated for its accuracy.

## 5. Conclusions

We found that polyp size, women’s BMI, and whether polyps appeared cystic or solid on ultrasound were helpful in a decision tree model to assess the risk of pre-malignancy or malignancy in postmenopausal polyps. In our study cohort, two-thirds of the women were classified as low-risk of cancer (≤5%), nearly one-third as intermediate-risk (>5% to ≤20%) and only 6% of women were considered high-risk (>20%). Our decision tree model may serve as a guide to aid the discussion between women and their clinicians on the benefits and risks of surgery to remove endometrial polyps. It may also help to prioritize women with a high risk of pre-malignancy or malignancy for surgery over those in whom the risk of malignancy is lower.

## Figures and Tables

**Figure 1 diagnostics-11-01094-f001:**
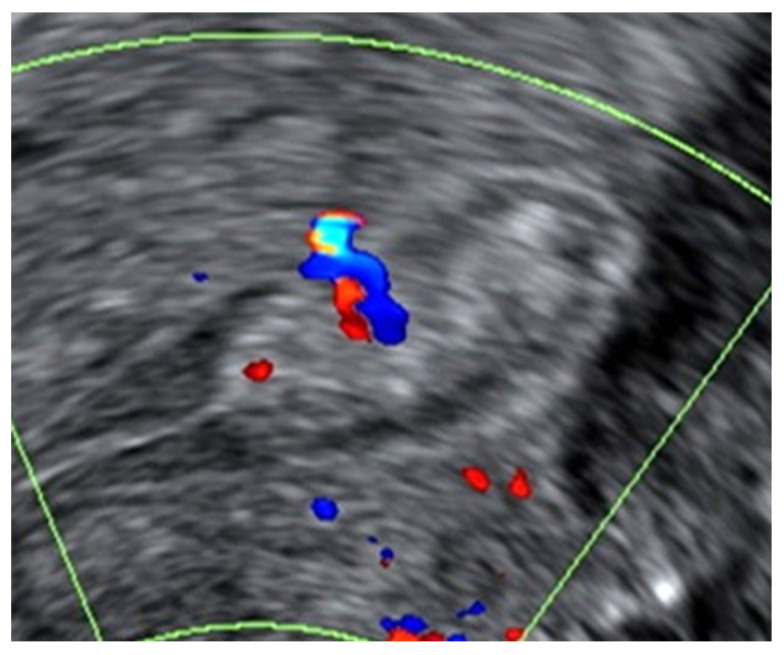
Example of a benign endometrial polyp with a regular outline and homogeneous echogenicity on B-mode greyscale ultrasound. On Doppler examination, the polyp had a single feeder vessel.

**Figure 2 diagnostics-11-01094-f002:**
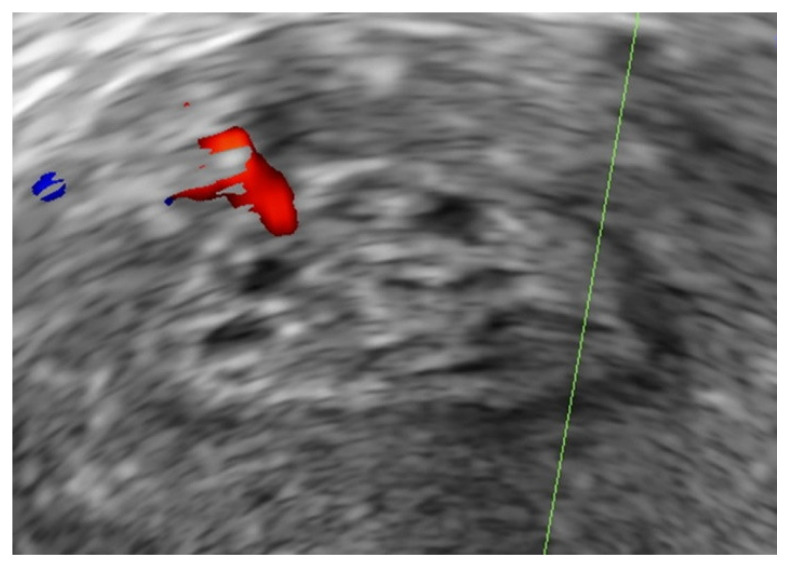
Example of an endometrial polyp with intralesional cystic spaces on B-mode greyscale ultrasound.

**Figure 3 diagnostics-11-01094-f003:**
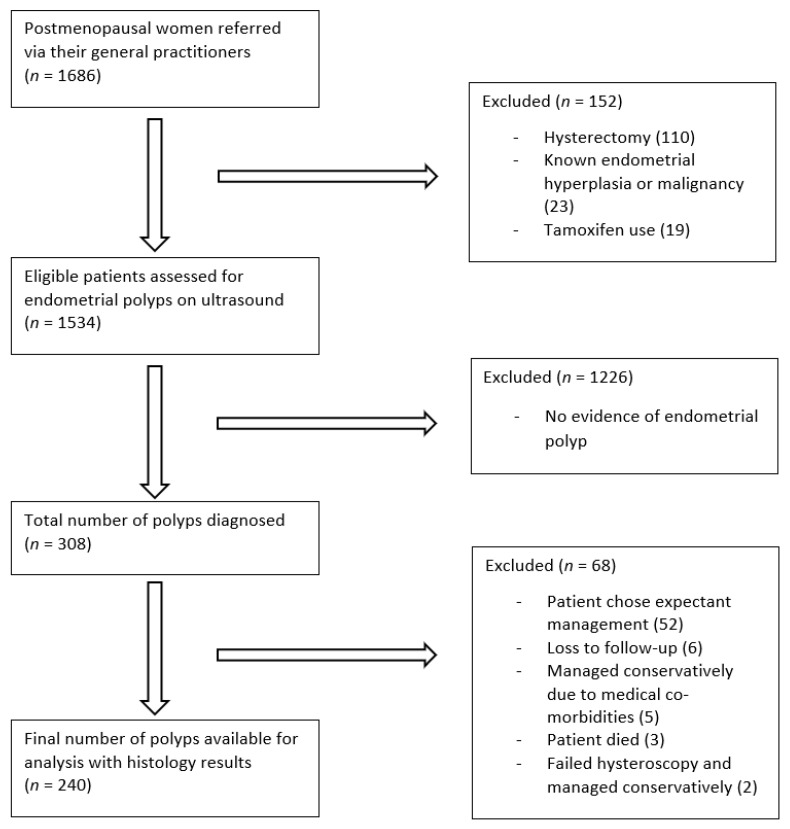
Study flow diagram.

**Figure 4 diagnostics-11-01094-f004:**
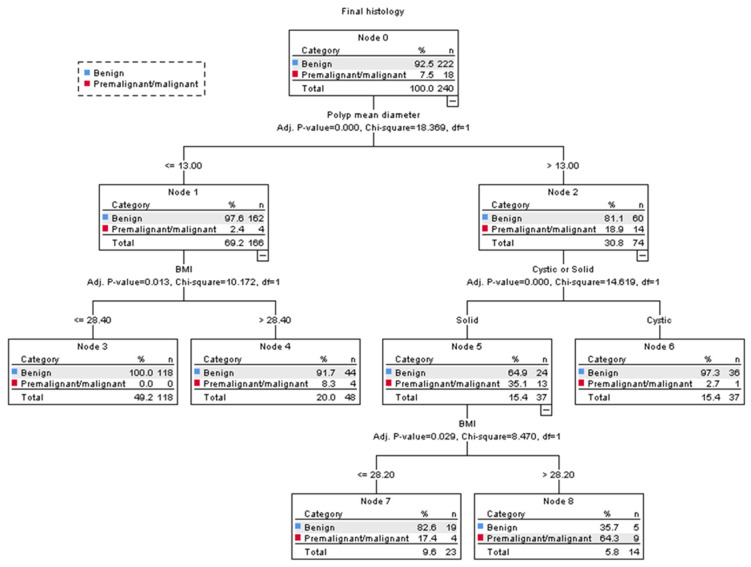
CHAID decision tree analysis on the risk of pre-malignancy or malignancy in women with postmenopausal polyps (Nodes 3 and 6 are classified as low-risk groups, 4 and 7 as intermediate-risk groups, and 8 as a high-risk group). The polyp mean diameter was measured in millimetres. BMI = body mass index.

**Table 1 diagnostics-11-01094-t001:** Indications for ultrasound examination (*n* = 1534).

Indications	*N* (%)
Postmenopausal bleeding	886 (57.8)
Abdominal or pelvic pain	209 (13.6)
Abdominal or pelvic swelling	146 (9.5)
Bowel or urinary symptoms	63 (4.1)
Incidental finding of a thickened endometrium on ultrasound	63 (4.1)
Raised serum CA125	52 (3.4)
Ovarian cancer screening	52 (3.4)
Other	63 (4.1)

**Table 2 diagnostics-11-01094-t002:** Patient characteristics of the study cohort (*n* = 240).

Characteristic	Benign Polyps (*n* = 222)	Pre-Malignant or Malignant Polyps (*n* = 18)
Age ^a^	60 (45–94)	65.5 (52–82)
BMI (kg/m^2^) ^b^	26.6 (18.5–52.3)	30.4 (21.6–40.8)
Nulliparity ^b^	63 (28.4)	6 (33.3)
Hypertension ^b^	75 (33.8)	10 (55.6)
Diabetes mellitus ^b^	22 (9.9)	3 (16.7)
Use of HRT ^b^	75 (33.8)	2 (11.1)
Symptoms of PMB ^b^	166 (74.8)	15 (83.3)

^a^ Median (range), ^b^ n (%), HRT = hormone replacement therapy, PMB = postmenopausal bleeding.

**Table 3 diagnostics-11-01094-t003:** Ultrasound morphological features of the endometrial polyps (*n* = 240).

Characteristic	Benign Polyps (*n* = 222)	Pre-Malignant or Malignant Polyps (*n* = 18)	Test Statistic	*p*-Value
Polyp mean diameter (mm) ^a^	10.0 (4.0–28.0)	13.3 (7.0–35.0)	U = 862.5	<0.001 ^c^
Presence of a pedicle vessel ^b^	92 (41.4)	12 (66.7)	n/a	0.048 ^d^
Presence of intralesional cystic spaces ^b^	82 (36.9)	1 (5.6)	n/a	0.008 ^d^
Multiple polyps ^b^	49 (22.1)	4 (22.2)	n/a	1.000 ^d^

^a^ Median (range), ^b^ n (%), ^c^ Mann–Whitney U test, ^d^ Fisher’s exact test, n/a = not applicable.

## Data Availability

The datasets used and analysed during the current study are available from the corresponding author on reasonable request.

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
