# Peer review of "Risk of Pre-Malignancy or Malignancy in Postmenopausal Endometrial Polyps: A CHAID Decision Tree Analysis"

_diagnostics, 2021, doi:10.3390/diagnostics11061094_

Round 1
Reviewer 1 Report
Congratulations on your excellent paper.
I wonder if you consider 3D US in your next research and office hysteroscopy for the polyp resection.
Author Response
Thank you very much for the reviewer's comments/suggestions.
Reviewer's comment/suggestion: I wonder if you consider 3D US in your next research and office hysteroscopy for the polyp resection.
Authors' response:
1. 3D ultrasound may provide a more accurate measurement of a polyp’s size compared to polyp's largest diameter or mean diameter on conventional ultrasound. However, to the best of our knowledge, polyp's volume on 3D ultrasound has not been used to predict the risk of pre-malignancy/malignancy in endometrial polyps. This could be due to the IETA consensus opinion (Leone 2009) recommending that “intracavitary lesions should be measured in three perpendicular diameters in millimeters, rounded up to one decimal point. And the volume of the lesion may be calculated from the three orthogonal diameters using the formula for a prolate ellipsoid”. Therefore, a clinical prediction model that uses polyp's volume on 3D ultrasound may not be as generalisable in routine clinical practice compared to the measurement of polyp's mean diameter on conventional ultrasound.
Furthermore, we analysed our results using both polyp's mean diameter and volume (using the formula for prolate ellipsoid) in our previous study on the natural history of endometrial polyps (Wong 2017), and we did not find that the results were significantly different when either method was used to measure polyp’s size.
2. Regarding office hysteroscopy for polyp resection, we routinely offer hysteroscopic polyp resection in the outpatient setting, only a small proportion of polyps are removed under general anaesthesia.
Interestingly, previous studies have reported that despite the normal appearance of a polyp during hysteroscopy, some polyps may still harbour foci of pre-malignancy/malignancy. For example, Fernandez-Parra et al. (2006) reported that hysteroscopy only had a sensitivity of 36% and specificity of 98% for premalignant/malignant polyps.
References:
LEONE, F. P. G., TIMMERMAN, D., BOURNE, T., VALENTIN, L., EPSTEIN, E., GOLDSTEIN, S. R., MARRET, H., PARSONS, A. K., GULL, B., ISTRE, O., SEPULVEDA, W., FERRAZZI, E. & VAN DEN BOSCH, T. 2010. Terms, definitions and measurements to describe the sonographic features of the endometrium and intrauterine lesions: a consensus opinion from the International Endometrial Tumor Analysis (IETA) group. Ultrasound in Obstetrics and Gynecology, 35, 103-112.
WONG, M., CRNOBRNJA, B., LIBERALE, V., DHARMARAJAH, K., WIDSCHWENDTER, M. & JURKOVIC, D. 2017. The natural history of endometrial polyps. Human Reproduction, 32, 340-345.
FERNÁNDEZ-PARRA, J., RODRÍGUEZ OLIVER, A., LÓPEZ CRIADO, S., PARRILLA FERNÁNDEZ, F. & MONTOYA VENTOSO, F. 2006. Hysteroscopic evaluation of endometrial polyps. International Journal of Gynecology & Obstetrics, 95, 144-148.
Reviewer 2 Report
The presented article is an interesting work, deserves the attention of readers. There are several questions: 1. Incorrectly classified samples must be analyzed to understand what could be the cause of the error. This information should be added. 2. Why, in your opinion, the values of tumor markers were not included in the decision tree?
Author Response
Thank you very much for the reviewer's comments/suggestions.
Reviewer's comment/suggestion: There are several questions: 1. Incorrectly classified samples must be analyzed to understand what could be the cause of the error. This information should be added. 2. Why, in your opinion, the values of tumor markers were not included in the decision tree?
Authors' response:
1. Our model recommends that polyp resection should be prioritised for women in the intermediate or high-risk groups, therefore only 1/18 premalignant/malignant polyp was incorrectly classified into the low-risk group. We have added an explanation of this false-negative diagnosis in our discussion (line 227-233). We concluded that although the presence of intralesional cystic spaces is useful in identifying benign polyps among larger-sized (>13mm) polyps, false-negative diagnoses can still occur in a small number of premalignant/malignant polyps that appear cystic rather than solid.
2. We acknowledge that serum tumour markers such as CA125 may be helpful in the detection of malignant endometrial polyps. However, in most of the early-staged endometrial cancers, the CA125 level can be normal. Furthermore, in our model, two-thirds of the polyps were classified as low-risk with a low false-negative rate, therefore the addition of serum tumour markers in this group of women is unlikely to improve the diagnostic accuracy further or be cost-effective. Nevertheless, in the high/intermediate-risk women, serum tumour markers and other imaging tests such as CT chest/abdomen/pelvis may have a role in predicting the presence of distal metastasis and aid clinical decision making.
